# ALIGNED TEXTUAL SCORING RULES

## ABSTRACT

Scoring rules elicit probabilistic predictions from a strategic agent by scoring the prediction against a ground truth state. A scoring rule is *proper* if, from the agent's perspective, reporting the true belief maximizes the expected score. With the development of language models, Wu & Hartline (2024) proposes a reduction from textual information elicitation to the numerical (i.e. probabilistic) information elicitation problem, which achieves provable properness for textual elicitation. However, not all proper scoring rules are well aligned with human preference over text. Our paper designs the Aligned Scoring rule (ASR) for text by optimizing and minimizing the mean squared error between a proper scoring rule and a reference score (e.g. human score). Our experiments show that our ASR outperforms previous methods in aligning with human preference while maintaining properness.

## 1 INTRODUCTION

The theory of proper scoring rules is well established for elicitation of numerical information, such as the probability of a random state (McCarthy, 1956; Savage, 1971), the mean of a distribution (Abernethy & Frongillo, 2012), and is widely used in practice (Danz et al., 2022; Hossain & Okui, 2013; Möbius et al., 2022). Proper scoring rules score the quality of a probabilistic prediction by comparing to the ground truth random state. By scoring a strategic agent, proper scoring rules are mechanisms that incentivize truthful prediction. Information Elicitation is an important area of research that has recent practical importance due to the reliance of data-driven algorithms and AI systems on high-quality input.

For example, in peer grading, students report predictions about their peers' homework correctness (the random state). The instructor spot-checks homework submissions and reveals the ground truth correctness. The student's prediction is then scored in comparison to the ground truth. A scoring rule is *proper* (a.k.a. truthful) if, from the peer's perspective, truthfully reporting her belief about the correctness maximizes expected score.

The recent development of large language models (LLM) has enabled the evaluation of textual information. Textual reports can encode richer information than numerical predictions. For peer grading, answering open-ended review questions facilitates the students' learning process better than checking pre-specified numerical rubrics. One approach to incentivize high-quality textual review from students is to score peer reviews by querying LLM to compare student reviews with the ground truth instructor review. Moreover, studies on language-model-generated evaluation systems, i.e., LLM-as-Judge (Zheng et al., 2023; Fu et al., 2024), have demonstrated that language models often align closely with human judgments when scoring text quality.

Language-model-generated evaluations offer scalability but, unfortunately, lack provable guarantees such as truthfulness, leaving them vulnerable to strategic manipulation. For example, when language models score peer reviews, fabricated comments may receive a higher expected score (Wu & Hartline, 2024). To address this issue, Wu & Hartline (2024) propose a reduction from textual elicitation problem to numerical elicitation problem. Wu & Hartline (2024) views a language model as an oracle accepting *summarization* and *question-answering* queries, where summarization identifies a scoring rubric with states for elicitation, and question-answering maps text to numerical reports and states. By implementing any numerical scoring rule over the identified space of rubrics, the scoring mechanism inherits provable properness when the language oracle is perfect and achieves adversarial robustness when the language oracle has errors. However, the scoring rules might not be aligned with preferences.

The goal of our paper is to align a provably proper textual proper scoring rule with preferences, e.g. human preferences. With the reduction framework in Wu & Hartline (2024), we optimize proper scoring rules to align with an exogenously given score that reflects a preference or a scoring rubric. For the peer grading application, we align the scoring rule to two reference scores: 1) the instructor score of peer reviews, and 2) the LLM-Judge score, by quering LLM to compare a peer review with the ground truth. While neither of these reference scores are proper, our optimization framework converts the reference scores into a proper score.

Our Aligned Scoring Rule (ASR) is simple, provably truthful, and interpretable. We minimize the Mean Squared Error (MSE) of ASR with the reference score. We optimize over the space of separate scoring rules, which applies a single-dimensional scoring rule to each summary point and averages across single-dimensional scores. The hypothesis space induces a convex optimization problem with efficient algorithms. The separate scoring rules allow us to interpret and identify the important rubric points from reference scores, by the convexity of each single-dimensional scoring rule.

We evaluate our Aligned Scoring Rule (ASR) on peer grading datasets. Results show that ASR fits the reference scores effectively and outperforms baselines. We first present the result of a linear regression that predicts the reference scores from ASR. The regression is nearly the identity function, showing our ASR aligns with reference scores. Then we present the MSE and the Pearson correlation between ASR and the reference score, in comparison with baseline methods including the best constant score and the method in Wu & Hartline (2024). Our ASR outperforms baseline methods in both metrics. Finally, we show the interpretability of ASR by a case demonstration in the appendix, where ASR identifies reasonably important and non-important rubric points for scoring.

## 1.1 RELATED WORK

**Textual Elicitation**    Several recent papers design scoring mechanisms to elicit textual information from language models. Kimpara et al. (2023) models LLM as a distribution that generates independent and identical (i.i.d.) textual samples. The paper designs a scoring rule that scores the distribution with access to samples, to incentivize a truthful report of the distribution, while our work directly scores the quality of a text. Lu et al. (2024) designs truthful peer prediction mechanisms that score text without ground truth, by comparing the textual report of multiple peers. Wu & Hartline (2024) designs proper scoring rules that score text with ground truth. The main goal of Lu et al. (2024); Wu & Hartline (2024) is truthfulness (a.k.a. properness), which does not consider optimization. On the contrary, our work optimizes over the space of proper scoring rules for alignment.

**Grading with LLMs**    Recent work studies the use of LLMs in grading textual reports from students. Kwiatkowski et al. (2019) studies grading via similarity between the vector embedding of the student report and ground truth. They show that the vector embedding approach works well for simple binary questions, but not for multiple-choice and more complex questions. Schneider et al. (2023) prompts a language model to compare student reports to ground truth, which is shown to have low Pearson correlation with instructor scores. Instead of directly prompting, our approach identifies scoring rubrics and optimizes for alignment while maintaining properness, thus having more favorable results.

**Automated Mechanism Design and Differentiable Economics**    Automated mechanism design (AMD) is the use of computational techniques to search for good mechanisms on specific problem instances. The earliest works in this area use linear programming (Conitzer & Sandholm, 2003a;b; Sandholm et al., 2007; Conitzer & Sandholm, 2004); others frame the problem in terms of learning theory, where the goal is to choose a high-performing mechanism from some class given access to samples from the type distribution (Roughgarden & Schrijvers, 2016; Morgenstern & Roughgarden, 2016; 2015; Balcan et al., 2008; Feldman et al., 2014; Hsu et al., 2016; Balcan et al., 2016; 2018b;a). A body of work sometimes called "differentiable economics" applies the tools of modern deep learning to learn good mechanisms, either using neural networks as general function approximators (Dütting et al., 2024), or using specially-designed architectures which guarantee strategyproofness in single-agent (Shen et al., 2019; Dütting et al., 2024; Curry et al., 2024) and multi-agent settings (Curry et al., 2022; Duan et al., 2023; Wang et al., 2024). Like early work on AMD, we solve a convex optimization to minimize expected loss from few samples. With more training data, applying differentiable economics' flexible function approximators is a promising future work.

**Optimization of Scoring Rules**   There is an extensive literature that characterizes proper scoring rules for numerical elicitation (McCarthy, 1956; Savage, 1971). Recently, a line of literature works on the optimization of scoring rules subject to normalization constraints such as boundedness. Li et al. (2022) optimizes to incentivize a binary effort in peer grading, where a peer either exerts effort to refine her posterior belief or not. As a generalization, Hartline et al. (2023) considers incentivizing a multi-dimensional effort. Our paper adopts the computation framework of the optimal scoring rule in Li et al. (2022). Additionally, Neyman et al. (2021) incentivizes sequential and discrete effort, Papireddygari & Waggoner (2022) connects proper scoring rules to contract theory, Chen & Yu (2021) considers robust scoring rule design that relaxes the knowledge of the prior of the designer, and Chen et al. (2023) designs optimal scoring rules in the online setting where the information structure and the cost of signals are unknown.

## 2   PRELIMINARIES

This section introduces the preliminaries of information elicitation and scoring rules we use.

### 2.1   NUMERICAL ELICITATION

The goal of the principal (mechanism designer) is to elicit numerical reports on the quality over $n$ explicit rubric points, represented by states $\boldsymbol{\theta} = (\theta_1, \ldots, \theta_n)$ where each $\theta_i \in [0, 1]$. The state space is $\Theta = [0, 1]^n$. For example, in peer grading, the rubric consists of Statement Correctness, Proof Correctness, and Clarity. A state being 1 means the highest quality on that rubric point. The agent holds a multi-dimensional belief $q \in \Delta([0, 1]^n)$ over the $n$ states. The principal asks the agent to report the marginal means $\boldsymbol{r} = (r_1, \ldots, r_n)$ from the report space $R = [0, 1]^n$.

The agent is scored by a scoring rule $S : R \times \Theta \to [0, 1]$ comparing the reported marginal means $\boldsymbol{r}$ and the realized state $\boldsymbol{\theta}$. A scoring rule is *proper* if the expected score is maximized when the agent reports the true marginal means of the state. From the agent's subjective perspective, the scoring rule incentivizes the agent to truthfully report the believed marginal means to maximize expected score.

**Definition 2.1** (Properness).   A scoring rule is *proper* for eliciting the marginal means, if for any belief distribution $q \in \Delta([0, 1]^n)$ with mean $\boldsymbol{\mu}_q$, and any deviation report $\boldsymbol{r} \in [0, 1]^n$,

$$\mathbf{E}_{\boldsymbol{\theta} \sim q}\left[S(\boldsymbol{\mu}_q, \boldsymbol{\theta})\right] \geq \mathbf{E}_{\boldsymbol{\theta} \sim q}\left[S(\boldsymbol{r}, \boldsymbol{\theta})\right].$$

A scoring rule is $\epsilon$-*approximately* proper if for any belief distribution $q \in \Delta([0, 1]^n)$ with mean $\boldsymbol{\mu}_q$, and any deviation report $\boldsymbol{r} \in [0, 1]^n$,

$$\mathbf{E}_{\boldsymbol{\theta} \sim q}\left[S(\boldsymbol{\mu}_q, \boldsymbol{\theta})\right] \geq \mathbf{E}_{\boldsymbol{\theta} \sim q}\left[S(\boldsymbol{r}, \boldsymbol{\theta})\right] - \epsilon.$$

Before reporting the belief, the agent holds a prior belief with marginal means $\boldsymbol{p} \in [0, 1]^n$, the empirical frequency of the ground truth in samples. The agent learns and refines the belief by receiving a signal $s \in S$ correlated with the ground truth state. The signal generation follows an information structure, a joint distribution $\Delta(\Theta \times S)$ over the state space and the signal space. Upon receiving the signal, the agent Bayesian updates to a posterior belief $q \in \Delta([0, 1]^n)$.

### 2.2   TEXTUAL ELICITATION

Text conveys implicit information rather than explicitly listed rubric points in numerical elicitation. Textual ground truth indicates a set of $m$ summary points. The reported summary points can be represented by an $m$-dimensional binary vector $\boldsymbol{\theta} = (\theta_1, \ldots, \theta_m)$, where $\theta_i \in \{0, 1\}$ for each $i$. State $\theta_i = 1$ or 0 means "agree" or "disagree" on the corresponding point. For example, in a peer review of an induction homework in an algorithm class, the summary points in the textual ground truth review contain $\theta_1$ the correctness of the hypothesis, $\theta_2$ the base case, and $\theta_3, \theta_4$ two details about some particular induction step. A reported text can express uncertainty on each state, e.g. "the base case is likely correct" as 70% probability that $\theta_2 = 1$ for base case.

In our peer grading dataset, we observe that textual reports either express a state being 0 or 1, or have no information. Thus, we restrict our attention to proper scoring rules with report space $r_i = \{0, 1, \perp\}$ for each $i$. We write $p_i$ as the empirical frequency of $\theta_i = 1$ in our dataset. Assumption 2.2 interprets an uncertain report of $\perp$ as the prior $p_i$.

**Assumption 2.2** (Know-it-or-not). In the peer grading dataset, the agent's posterior belief distribution $q_i$ is either 0, 1, or the prior $p_i$.

Assumption 2.2 restricts the space of proper scoring rules to scoring rules for report space $R = \{0, 1, \perp\}$.

**Definition 2.3** (Scoring Rules for Know-it-or-not Reports). Given the prior distributions $\boldsymbol{p}$, a scoring rule $S_{\boldsymbol{p}} : \{0, 1, \perp\}^m \times \{0, 1, \perp\}^m \to [0, 1]$ for know-it-or-not reports is proper if there exists a proper scoring rule $S : [0, 1]^m \times \{0, 1\}^m \to [0, 1]$, such that

$$S_p(\boldsymbol{r}, \boldsymbol{\theta}) = S(\tilde{r}_{\boldsymbol{p}}(\boldsymbol{r}), \boldsymbol{\theta}),$$

where $\tilde{r}_{\boldsymbol{p}}$ maps a report to the probabilistic belief, particularly, $\perp$ to the prior:

$$\tilde{r}_{\boldsymbol{p}}(r_i) = \begin{cases} r_i & \text{if } r_i \in \{0, 1\} \\ p_i & \text{else, when } r_i = \perp. \end{cases}$$

A scoring rule for multi-dimensional summary points can be defined from single-dimensional scoring rules and multi-dimensional aggregations.

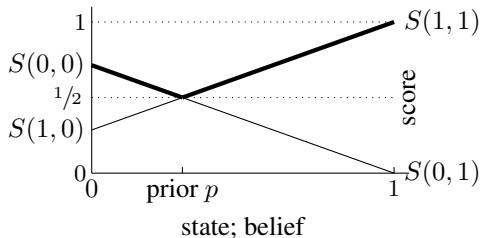

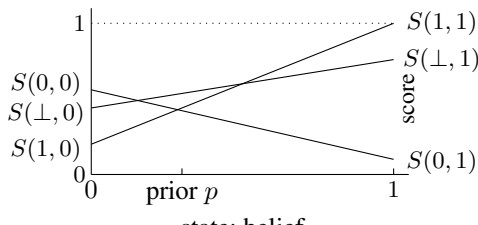

Figure 1: The V-shaped scoring rule, the optimal scoring rule in Li et al. (2022). Once fixing a report, the expected score is a linear line in both the realized state and the mean of the ground truth distribution. Reporting the prior always gets a score of $1/2$ (the dotted line). The V-shaped upper envelope of the two linear lines forms the expected score of a truthful agent.

Figure 2: An example of a single-dimensional scoring rule for know-it-or-not reports. Each report in the ternary space corresponds to a linear line. The scoring rule can be depicted by three linear lines. Properness requires that, when the belief (or, equivalently, the ground truth) is $r$, the line with the highest expected score is on the line corresponding to report $r$.

**Single-Dimensional Scoring Rule** We introduce the V-shaped scoring rule and the single-dimensional scoring rule for know-it-or-not reports here.

The V-shaped scoring rule is introduced in Li et al. (2022) as the optimal scoring rule that incentivizes a binary effort, when the agent can choose to exert effort and update her belief from prior to posterior. Wu & Hartline (2024) tests aggregations over the V-shaped scoring rule. The V-shaped scoring rule partitions the report space into a ternary space: a report higher than the prior mean, lower than the prior mean, or the prior mean $p$. Figure 1 depicts a V-shaped scoring rule with $p < \frac{1}{2}$.

**Definition 2.4** (V-shaped Scoring Rule). Given prior mean $p \in [0, 1]$, a V-shaped scoring rule $S : [0, 1] \times [0, 1] \to [0, \frac{1}{2}]$ is defined by

$$S_p(r, \theta) = \begin{cases} \frac{1}{2} - \frac{1}{2} \cdot \frac{\theta - p}{1 - p} & \text{if } r < p \\ \frac{1}{2} + \frac{1}{2} \cdot \frac{\theta - p}{1 - p} & \text{if } r > p \\ \frac{1}{2} & \text{else} \end{cases}$$

When $p \in (\frac{1}{2}, 1]$, the score is symmetric, i.e. $S_p(r, \theta) = S_{1-p}(1 - r, 1 - \theta)$.

A single-dimensional scoring rule for know-it-or-not reports can be characterized by nine values: $S(r, \theta)$ for $r \in \{0, 1, \perp\}$ and $\theta \in \{0, 1\}$. The definition of properness simplifies to Definition 2.5. A V-shaped scoring rule is a special case of a single-dimensional scoring rule for know-it-or-not reports, where the score of reporting $\perp$ is fixed at $\frac{1}{2}$. Figure 2 presents a graphical illustration of such a scoring rule.

**Definition 2.5.** With prior $p$, a single-dimensional scoring rule for know-it-or-not reports is proper if

$$S(\theta, \theta) \geq S(r, \theta), \qquad \forall \theta \in \{0, 1\}, \forall r \in \{0, 1, \bot\}$$

$$\mathbf{E}_{\theta \sim p}\left[S(\bot, \theta)\right] \geq \mathbf{E}_{\theta \sim p}\left[S(r, \theta)\right], \qquad \forall r \in \{0, 1, \bot\}$$

**Multi-Dimensional Aggregations** A multi-dimensional aggregation operates over single dimensional scoring rules and preserves properness.

**Definition 2.6.** Given single dimensional scoring rules $S_1, \ldots, S_m$ where each $S_i : [0, 1] \times [0, 1] \to [0, 1]$, a multi-dimensional scoring rule $S : [0, 1]^m \times [0, 1]^m \to [0, 1]$ is aggregated from $S_1, \ldots, S_m$ if 1) $S$ is proper, and 2) there exists aggregation function $A$ such that

$$S(r_1, \ldots, r_n; \cdot) = A\big(S_1(r_1; \cdot), \ldots, S_n(r_n; \cdot)\big).$$

We introduce two aggregations, the separate aggregation and the max-over-separate (M) aggregation.

We optimize over the space of separate scoring rules (Li et al., 2022). Wu & Hartline (2024) also tests the averaged V-shaped scoring rule (AV).

**Definition 2.7.** Given scoring rules $S_1, \ldots, S_m$, a separate scoring rule is the weighed average $S = \sum_{i \in [m]} w_i S_i$, with weights $w_1, \ldots, w_m$ such that $\sum_{i \in [m]} w_i = 1$.

The max-over-separate scoring rule scores an agent by the dimension on which the agent has the highest expected score. It can be implemented by asking the agent to pick her favorite dimension and score on that dimension. Wu & Hartline (2024) tests the max-over-separate V-shaped scoring rule (MV), the optimal scoring rule in the multi-dimensional report. We will compare our Aligned Scoring Rule with the MV scoring rule.

**Definition 2.8** (Max-Over-Separate). Given scoring rules $S_1, \ldots, S_m$, a max-over-separate scoring rule is

$$S(\boldsymbol{r}, \boldsymbol{\theta}) = S_i(r_i, \theta_i), \text{ where } i = \arg\max_{i'} \mathbf{E}_{\theta_{i'}}\left[S_{i'}(r_{i'}, \theta_{i'})\right].$$

## 3 ALIGNED SCORING RULE: ALGORITHM

In this section, we present our design of Aligned Scoring Rule (ASR), which reduces textual elicitation to numerical elicitation and optimizes for human alignment in peer grading. Section 3.1 list the provable properness guarantees of the reduction from Wu & Hartline (2024). Section 3.2 describes our optimization method for alignment.

Following Wu & Hartline (2024), we model the language model as an oracle accepting *Summarization* and *Question-Answering* queries, which are fundamental natural language processing tasks (Bar-Haim et al., 2020; Clark et al., 2019; Rajpurkar et al., 2016). The Summarization oracle outputs a list of summary points from a list of texts. The Question-Answering oracle identifies whether a text agrees or disagrees with a summary point.

**Summarization** $O_S$, summarizes a list of textual report into summary points.

    **Input** A list of texts $\mathtt{T}_1, \ldots, \mathtt{T}_N$.

    **Output** A list $[\mathtt{t}_1, \ldots, \mathtt{t}_m]$ of all summary points from texts.

**Question-Answering** $O_A$ determines whether a text agrees or disagrees with a summary point, or is not applicable.

    **Input** One text $\mathtt{T}$ and a summary point $\mathtt{t}$.

    **Output** Output "disagree" 0, "agree" 1, or "NA" $\bot$.

We describe Elicitation$^{\text{GPT}}$ from Wu & Hartline (2024) here. Following Assumption 2.2, we map a report $\bot$ to the prior report, the empirical frequency of a summary point. The clustered nature of the peer grading application enables the identification of the empirical frequency. The dataset is partitioned in advance into clusters. Each cluster contains $N$ peer grading tasks, where the homework submission are all from the same assignment, thus applicable to the same set of grading rubrics.

**Input**  $N$ ground truth reviews $\{\mathtt{I}_1, \ldots, \mathtt{I}_N\}$ on submissions to the same homework assignment; one reported review $\mathtt{R}_k$ on the $k$th submission; and a proper scoring rule $S$ for know-it-or-know beliefs. We will write the identified states and reports by the language oracle as $\hat{\boldsymbol{\theta}}$ and $\hat{\boldsymbol{r}}$, respectively.

**Algorithm (Elicitation$^{\mathbf{GPT}}$)**

- (Summarization) Summarize instructor reviews into points. $\{\mathtt{t}_1, \ldots, \mathtt{t}_m\} = O_S(\{\mathtt{I}_i\}_{i \in [N]})$.

- (Question-Answering) Map truth $\mathtt{I}_i$ to state space. For each instructor review $j \in [N]$ and each summary point $i \in [m]$, $\theta_i^j = O_A(\mathtt{I}_j, \mathtt{t}_i)$. Calculate the prior of each state $p_i = \frac{1}{N}\sum_j \theta_i^j$.

- (Question-Answering) Map the review to report space. For each point $i \in [m]$, $\hat{r}_i = O_A(\mathtt{R}_k, \mathtt{t}_i)$.

- Apply proper scoring rule for know-it-or-not reports. Output $S_{\boldsymbol{p}}(\boldsymbol{r}, \boldsymbol{\theta}^k)$[1].

### 3.1   PROVABLE PROPERNESS

We list the provable property of the reduction here, including the case that the language oracle makes errors and adversarial robustness.

The correctness of summarization $O_S$ does not affect the truthfulness of Elicitation$^{\mathrm{GPT}}$. To see this, even when $O_S$ misidentifies the summary points, Elicitation$^{\mathrm{GPT}}$ is still proper as long as $O_A$ correctly identifies the numerical states and reports corresponding to the summaries. We assume $O_A$ is perfect on the ground truth side, as the ground truth reviews often clearly state opinions on summary points.

When the language oracle $O_A$ is non-inverting on the report side, Elicitation$^{\mathrm{GPT}}$ is proper.

**Definition 3.1** (Non-inverting $O_A$). The question-answering oracle for know-it-or-not beliefs is non-inverting if the probability of inverting the report is strictly less than $\frac{1}{2}$, i.e. $\Pr[\hat{r}_i \neq r_i | \mathtt{R}] \leq \frac{1}{2}$ for any $i$ and any $\mathtt{R}$.

**Theorem 3.2** (Wu & Hartline 2024). *If the question-answering oracle for know-it-or-not beliefs is non-inverting for reports, Elicitation$^{GPT}$ is proper.*

Without assumptions on the language oracle's error, the reduction above has adversarial robustness.

**Theorem 3.3** (Wu & Hartline 2024). *If the agent has no information, the highest expected score she can achieve is at most by saying $\perp$ (i.e. "I don't know").*

### 3.2   OPTIMIZATION FOR ALIGNMENT

While Elicitation$^{\mathrm{GPT}}$ presents a framework for reducing textual elicitation to numerical elicitation, not all proper scoring rules align well with the instructor preferences. Thus, our Aligned Scoring Rule (ASR) optimizes over a space of separate scoring rules and selects the one that aligns best with the reference score, i.e., the instructor score of a peer review. Our optimization framework follows the computation of optimal scoring rule in Li et al. (2022). Our Aligned scoring rule can be viewed as a truthful proxy of the instructor score.

Fixing summary points $\{\mathtt{t}_1, \ldots, \mathtt{t}_m\}$ and prior $\boldsymbol{p}$, our optimization objective minimizes the mean squared error (MSE) between Elicitation$^{\mathrm{GPT}}$ score and the reference score (e.g. instructor score). Our optimization problem is shown in Program 1 with $s$ normalized to $[0, 1]$.

$$\min_{\{S\}_{i \in [m]}} \quad \mathbf{E}_{(\boldsymbol{r}, \boldsymbol{\theta}, s)} \left[ (S(\boldsymbol{r}, \boldsymbol{\theta}) - s)^2 \right] \tag{1}$$
$$\text{s.t.} \quad S \text{ is proper}$$
$$S(\cdot, \cdot) \in [0, 1]$$

We optimize over the space of separate scoring rules, the sum of single-dimensional proper scoring rules $\{S_i\}_{i \in [m]}$ for know-it-or-not reports. A separate scoring rule is simple and interpretable, where the convexity of single-dimensional scores can identify the importance of each dimension. Program

---

[1]Note that the ground truth may have $\perp$ in our implementation. In such a case, we score the student by the expected score where the binary state is drawn from the prior.

2 shows the simplified optimization problem for separate scoring rules. The properness constraint follows properness for know-it-or-not reports in Definition 2.5.

$$\min_{\{S_i\}_{i\in[m]}} \mathbf{E}_{(\boldsymbol{r},\boldsymbol{\theta},s)}\left[\left(\sum_{i\in[m]} S_i(r_i,\theta_i) - s\right)^2\right] \tag{2}$$

$$\text{s.t. for any dimension } i, \tag{Properness}$$
$$\text{for any } r_i \in \{0,1,\perp\}$$
$$S_i(\theta_i,\theta_i) \geq S_i(r_i,\theta_i), \forall \theta_i \in \{0,1\}$$
$$\mathbf{E}_{\theta_i\sim p_i}\left[S_i(\perp,\theta_i)\right] \geq \mathbf{E}_{\theta_i\sim p_i}\left[S_i(r_i,\theta_i)\right]$$
$$\sum_{i\in[m]} S_i(r_i,\theta_i) \in [0,1], \forall \boldsymbol{r},\boldsymbol{\theta} \tag{Boundedness}$$

Our optimization problem with separate scoring rules is convex. Note that this formulation may not be convex for other spaces of multi-dimensional scoring rules, e.g. max-over-separate scoring rules.

**Corollary 3.4.** *Optimization problem 2 is convex.*

To see Corollary 3.4, note that for each dimension, we have six variables: $S_i(r_i,\theta_i)$ for $r_i \in \{0,1,\perp\}$ and $\theta_i \in \{0,1\}$. Both our objective and constraints are convex in the variables. Since optimization problem 2 is convex, we optimize with the gradient descent algorithm over samples.

## 4 IMPLEMENTATION OF LANGUAGE ORACLES

We describe our implementation of the language oracle here.

### 4.1 SUMMARIZATION ORACLE

The implementation of the summarization oracle includes three steps: summarizing instructor reviews, preparing negative/positive statement pairs from reviews, and clustering negative/positive statement pairs. Note that instead of directly clustering summary statements by similar meanings, for each statement from the reviews, we concatenate the statement with another of the opposite meaning to prepare a pair of negative/positive statements. The negative/positive statement pairs improve the robustness of LLM clustering. When each summary point consists of negative/positive statement pairs, the semantic meaning of each state can be viewed as neutral, avoiding opposite statements being identified as different states for elicitation.

**Input**  A list of $N$ instructor reviews $[\mathtt{I}_1, \ldots, \mathtt{I}_N]$.

**Output**  A list $[\mathtt{t}_j]_{j\in m}$ of summary points from reviews.

**Implementation**  We provide a toy prompt with each step below. The real prompts we use are listed in Appendix A.

- Summarize each instructor review into summary points.
  *Toy prompt: Carefully read the entire review comment. Extract all evaluative statements from the review. These should be comments that assess the quality, strengths, weaknesses, and suggestions. Ignore purely descriptive statements. Create an indexed list of these evaluative statements.*
- Transform each statement into negative/positive pairs.
  *Toy prompt: You are tasked with creating opposite evaluative statements for a given list of evaluative statements. For each statement provided, you need to create a new statement that has the same content but expresses the opposite emotion or sentiment.*
- Cluster the negative/positive pairs of summary points. The semantic meaning of each cluster is identified as the dimension for elicitation, $[\mathtt{t}_j]_{j\in[m]}$.
  *Toy prompt: You will be given a list of opinion pairs, each with a positive and corresponding negative opinion. Your task is to analyze these pairs and cluster them based on similarity.*

## 4.2 QUESTION-ANSWERING ORACLE

We directly query LLM to identify whether a review `R` is positive or negative for a summary point `t`.

**Input**    One review `R` and a summary point `t`.

**Output**    Positive (1), negative (0), or NA ($\perp$).

**Implementation** We provide an toy prompt below. The real prompt we use are listed in Appendix A.

***Toy prompt****: Your task is to infer which of the given positive/negative opinions is correct based on the provided review comment. For each opinion pair, read and understand both the positive and negative opinions. Conclude whether the review supports the positive, the negative, or neither opinion.*

## 5 EMPIRICAL EVALUATION

We describe our dataset and evaluation metric in Section 5.1, our reference scores used for alignment in Section 5.2, and our experimental results in Section 5.3. We depict the Aligned Scoring Rule (ASR) for one example homework assignment in Appendix C.

### 5.1 DATASET AND EVALUATION METRIC

**Dataset**    We present results from peer grading data in two undergraduate algorithm classes. Our dataset includes 22 assignments in total.[2] Each assignment has 6 to 8 homework submissions. Each homework submission has one instructor review (i.e. ground truth) and 6 to 8 peer reviews. Each peer review has an instructor score in $[0, 10]$.

**Metric**    We report the *Mean Squared Error*, the *Pearson correlation coefficient*, and the *Spearman rank correlation coefficient* of our ASR compared with reference scores.

- MSE quantifies the average magnitude of prediction errors.
- Pearson correlation assesses the strength of the linear relationship between predicted scores and reference scores, capturing whether the model correctly preserves the relative ordering.
- Spearman rank correlation assesses the correlation between two ranks.

### 5.2 REFERENCE SCORE

We optimize for alignment with two reference scores, the Instructor Score and the LLM-Judge Score.

**Instructor Score**    Instructor score (i.e., human preference) from our dataset.

**LLM-Judge Score**    We query a language model to grade the peer review against the instructor review based on a given peer review scoring rubric.

There is a high correlation between the Instructor Score and LLM-Judge score. Figure 3 presents the empirical joint distribution of Instructor Score and LLM-Judge Score for all data, with a Pearson correlation of 0.5540. The results show that LLM-Judge score can serve as a substitute for the costly and noisy instructor score, improving the scalability and the robustness of the peer grading system, which is consistent with previous studies of the LLM-as-Judge method, e.g., Zheng et al., 2023; Hackl et al., 2023, etc.

Note, the instructor and LLM-judge reference scores are not proper and therefore might encourage peer reviewers to engage in strategic behavior like guessing or adding irrelevant statements (Wu & Hartline, 2024). Our method of aligning a proper scoring rule to these references can be viewed as converting these non-proper scores into proper ones.

---

[2]Algorithm Class 1: 276 reviews by 23 peers on 89 submissions across 12 assignments. Algorithm Class 2: 240 reviews by 24 peers on 59 submissions across 10 assignments.

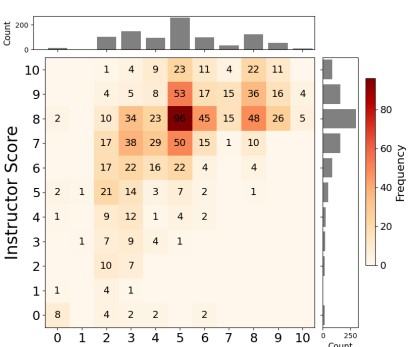

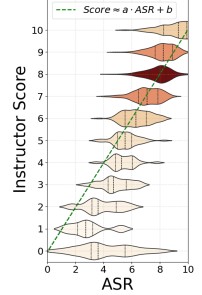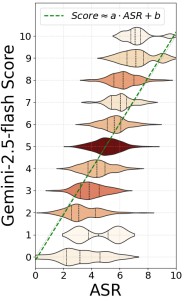

Figure 4: Reference Scores vs. ASR. Left: instructor score vs. ASR aligned with instructor score. Right: LLM-Judge score vs. ASR aligned with LLM-Judge score. The green line represents the linear regression fitting reference score from ASR, which is nearly the identity function in both plots.

Figure 3: Joint distribution (instructor score vs. LLM-Judge score).

## 5.3 EXPERIMENTAL RESULTS

We present our experimental results in this section. First, we show that a linear regression fitting the reference score from our ASR results in a nearly-identity linear fit. We then present the MSE and the correlation coefficients and compare with baselines. We use the Gemini-2.5 series models for the LLM-Judge and the language oracles and provide experimental details in Appendix A. We also tested the performance of GPT-4.1 as the LLM-Judge, with the results detailed in Appendix B.

**Nearly-Identity Linear Fit**  The first criterion for evaluating our approach is to examine whether our ASR can effectively fit the original reference scores. Figure 4 illustrates the joint empirical distribution of the ASR scores and the reference scores, with a regression line predicting the reference score $s$ from the ASR score $S$. The parameters of linear regression align closely with $s = S$.

| (a) Reference: Instructor Score | | | | (b) Reference: LLM-Judge Score | | | |
|---|---|---|---|---|---|---|---|
| Method | SquaredLoss | PearsonCorr | SpearmanCorr | Method | SquaredLoss | PearsonCorr | SpearmanCorr |
| ASR | 1.730 | 0.717 | 0.622 | ASR | 2.003 | 0.705 | 0.658 |
| Constant | 3.741 | N/A | N/A | Constant | 4.136 | N/A | N/A |
| EGPT(AV) | 9.541 | 0.294 | 0.301 | EGPT(AV) | 7.053 | 0.328 | 0.338 |
| EGPT(MV) | 18.360 | 0.213 | 0.207 | EGPT(MV) | 17.069 | 0.246 | 0.226 |

Table 1: Comparison with baselines.

**Comparison with Baselines**  Our Aligned Scoring Rule is compared against the following two baselines which are all truthful:

1. **Best Constant Score** ($S_{\text{const}}$). This method outputs the best constant score for all reviews, which is the mean of the reference scores $s$ in the training data $D$. The constant score is weakly truthful.

$$S_{\text{const}}(r_{\text{T}}, \theta_{\text{T}}) = \sum\nolimits_{(r,\theta,s)\in D} s/|D|.$$

2. **Non-aligned ElicitationGPT (EGPT)**. We compare with the Elicitation[GPT] in Wu & Hartline (2024), which is not aligned to a reference, particularly, the averaged V-shaped scoring rule (AV) and the max-over-separate V-shaped scoring rule (MV). In Wu & Hartline (2024), the AV scoring rule is shown to align the best with instructor score. Note that the max-over-separate scoring rule is not in our hypothesis space of separate scoring rules, and does not induce a convex optimization problem. [3]

The performance of scores is evaluated along three metrics: MSE, the Pearson correlation coefficient, and the Spearman rank correlation coefficient. Our ASR aligns best with the reference on all metrics.

---

[3]We evaluate Spearman correlation differently from Wu & Hartline (2024). They evaluate the ranking of the same student's averaged scores over all peer reviews in a class, because the Elicitation[GPT] scores are not in the same scale as reference scores. We evaluate each individual peer review's ranking, as our score is aligned.

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

# A   IMPLEMENTATION DETAILS

In this section, we provide a detailed description of how we implement our methods and conduct the experiments, including the prompts and other parameters for LLM calls, the numerical solution to the convex optimization problem, as well as the pre/post-processing of human feedback.

## A.1   LLM CALLS

We use the gemini-2.5 series models as the LLM oracles in our experiments. Specifically, we experiment with gemini-2.5-flash-preview-04-17 for all tasks other than clustering the negative/potitive pairs. For clustering, we employed gemini-2.5-pro-preview-05-06 due to its proficiency in handling long contexts. While calling LLMs, we set the temperature to 0, the "thinking" feature disabled, and maximum output token 8192. Next, we will provide a detailed description of each prompt used.

### A.1.1   SUMMARIZATION ORACLE

The implementation of the summarization oracle includes three steps: summarizing instructor review, preparing negative/positive statement pairs from reviews, and clustering negative/positive statement pairs.

---

**Summarizing Instructor Review**

You are an AI assistant specializing in analyzing assignment reviews. Your task is to extract all evaluative points from a given review comment.

<review_comment>REVIEW_COMMENT</review_comment>

Please follow these steps to analyze the review comment:

1. Carefully read the entire review comment.

2. Extract all evaluative statements from the review. These should be comments that assess the quality, strengths, weaknesses, and suggestions. Ignore purely descriptive or meaningless statements. Ignore statements purely about specific scores and ratings.

3. Create an indexed list of these evaluative statements. Each entry should be a single sentence in a single line containing a distinct evaluation from the review.

- You should clearly convey the sentiment behind an evaluative statement.

4. After creating the indexed list. Split and Rewrite each evaluative statement into several abstract and concise statements, abandoning the specific expression.

- Make your entry abstract and concise.

- Always use "part A / B / C" in the output to refer parts, even if the input says "part a / b / c" or "part 1 / 2 / 3".

- If an evaluative statement contains comments on multiple distinct aspects, they need to be listed as multiple entries.

Example: "I like the overall idea, but authors need to revise the presentation and experiments" have 3 different aspects, "The overall idea is good", "The presentation need revision", and "The experiments need revision".

Example: "Part A is correct and part B is wrong" have 2 different aspects, "Part A is correct", and "Part B is wrong".

- Ignore the unimportant positive parts of negative statements and the unimportant negative parts of positive statements.

- Each new entry inherits the index of the original entry, even if there are duplicate indexes.

Your output should be structured as follows:

<numbered_entries>[List your numbered entries here, one per line]</numbered_entries>

<rewrited_entries>[Rewrite each entry into an abstract and concise statement]</rewrited_entries>

---

**Preparing Negative/Positive Statement Pair**

You are tasked with creating opposite evaluative statements for a given list of evaluative statements. For each statement provided, you need to create a new statement that has the same content but expresses the opposite emotion or sentiment.

In addition, you also need to output whether the sentiment of the original statement is positive or negative.

Guidelines for creating opposite evaluative statements:

1. Maintain the same subject matter and key elements of the original statement.

2. Change the emotional tone or sentiment to its opposite (e.g., positive to negative, approval to disapproval).

3. Use similar language structure when possible, but modify words to reflect the opposite sentiment.

4. Ensure the new statement is coherent and makes sense in isolation.

5. Make the new statement as concise as possible.

Here is the list of evaluative statements:

<evaluative_statements>

EVALUATIVE_STATEMENTS

</evaluative_statements>

For each statement in the list, create an opposite version following the guidelines above. Present your results in the following format:

<result_1>

<original>[Original evaluative statement]</original>

<sentiment>[Sentiment of the original evaluative statement]</sentiment>

<opposite>[Your created opposite evaluative statement]</opposite>

</result_1>

<result_2>

...

</result_2>

...

Ensure that each opposite statement accurately reflects a reversal of sentiment while maintaining the core content of the original statement.

**Clustering Statement Pairs**

You will be given a list of opinion pairs, where each pair consists of a positive opinion and its corresponding negative opinion. Your task is to analyze these pairs and cluster them based on similarity. Follow these steps:

1. First, read the list of opinion pairs provided:

<opinion_pairs>OPINION_PAIRS</opinion_pairs>

2. Next, cluster the unique pairs based on their similarity in topic or theme in <clustering> tag. Pairs in the same cluster should address roughly the same aspects of the subject matter. Follow these steps:

1) You need to first draft a set of cluster descriptions in the <draft> tag. Each cluster description must be specific:

- You should cluster opinion pairs discussing different parts in different clusters.

- The description should clearly indicate the target of evaluation, avoiding terms like "overall" or "assignment" and instead using "the proof," "part A," or "the answer."

- The description should clearly specify the evaluation criteria, avoiding terms like "quality" and instead using "correctness," "clarity," or "detail."

2) Then, based on these descriptions, analyze the following aspects in the <analysis> tag:

- Splitting and merging clusters: Merge clusters that are redundant. Split clusters that contain more than one parts or aspects.

- New clusters: Look for opinions that are not covered by any existing cluster. Create a new cluster when at least two opinions fit it, and ignore any lone opinion that cannot be grouped.

- Specificity check: Ensure each cluster description includes specific evaluation criteria, rather than vague terms.

- Limit the number of clusters: Ensure the total number of clusters is between 10 and 12.

3) After completing this analysis, redefine the cluster descriptions based on your findings and repeat the entire process.

4) Perform this iteration a total of four times, wrapping the results of each iteration inside <epoch_i> tags, where i represents the iteration number.

You should follow this output format:

<clustering>
<epoch_1>
<draft>[Your draft cluster descriptions]</draft>
<analysis>[Your analysis here]</analysis>
</epoch_1>
<epoch_2>
...
</epoch_2>
...
</clustering>

3. Complete your final cluster descriptions. For each cluster, generate an opinion pair as the cluster representative.

- Ensure the opinion pair discusses exactly the core idea of the cluster description.

- The opinion pair should be brief and omit details.

- Do not use "need" or "need not" in your opinion pair. Instead, express what was done or what was failed to be done.

- Ensure the positive opinion and the negative opinion present exact opposing views.

- It is not necessary to summarize all content. Focus only on evaluating the most important aspect, and avoid using "and" to connect different aspects.

- Avoid using extreme words such as "excellent" and "awful."

You should follow this output format:

<clusters>
<cluster_1>
<description>[The description of the cluster]</description>
<representative>[Positive opinion] [Negative opinion]</representative>
</cluster_1>
<cluster_2>
...
</cluster_2>
...
</clusters>

### A.1.2 QUESTION-ANSWERING ORACLE

We directly query LLM to identify whether the review R is positive or negative for the summary point t.

**Input** One review R and a summary point t.

**Output** Positive (1), negative (0), or NA ($\perp$).

**Question-Answering Oracle**

You are an AI assistant specializing in analyzing assignment reviews. Your task is to infer which of the given positive/negative opinions is correct based on the provided review comment. You will be given two inputs:

<review_comment>REVIEW_COMMENT</review_comment>

<opinion_pairs>OPINION_PAIRS</opinion_pairs>

The review comment is the text of the review that you need to analyze. The opinion pairs consist of several lines, each containing a positive evaluation and its corresponding negative evaluation.

For each opinion pair, follow these steps to analyze and conclude in <result> tag:

1. Reprint the index of the opinion pair in <index> tag.

2. Copy the text of the opinion pair in <opinion_pair> tag.

3. Carefully read and understand both the positive and negative opinions.

4. List all possibly relevant statements in the comment one by one in the <statements> tag. For each relevant statement, determine whether it supports the positive opinion, the negative opinion, or neither, and specify whether the support is explicit or partial.

- Focus on the original meaning of the statement and avoid speculation as much as possible.

Example: The correctness of the assignment refers to the accuracy of the final answer and does not include the reasoning process.

Example: The correctness of the proof / claim does not affect the correctness of the answer.

Example: The wrong proof / answer / reasoning does not affect clarity.

5. Apply the following rules to determine the final conclusion in the <rubric> tag:

- If only one direction is supported, classify as that direction, even if it is only partially supported.

- If their are conflicts, classify as the direction with stronger support.

- If no statement is relevant to the opinion pair, classify as "Neither". Avoid selecting "Neither" whenever possible.

- At the end of the rubric, explicitly state you choose "Positive", "Negative", or "Neither".

6. Restate your choice of whether the review supports the positive, the negative, or neither in the <conclusion> tag.

- Only contain "Positive", "Negative", or "Neither" in the tag! Do not use words like "Correct", "Incorrect", "Clear", "Unclear".

Present your analysis and conclusion for each opinion pair in the following format:

<result>

<index>[The index of the input opinion pair here]</index>

<opinion_pair>[Copy the input opinion pair here]</opinion_pair>

<statements>

Statement: [Statement 1]

Analysis: [Analysis for Statement 1]

Statement: [Statement 2]

Analysis: [Analysis for Statement 2]

...

</statement>

<rubric>[Apply the rubric here]</rubric>

<conclusion>[Positive / Negative / Neither]</conclusion>

</result>

<result>...</result>

...

### A.1.3 LLM SCORE

**LLM Score**

You are an AI assistant specializing in educational assessment. Your task is to evaluate a peer review of a course assignment by comparing it to an instructor's review of the same assignment. You will analyze the alignment between the two reviews and assign a score from 0 to 10.

First, you will be given the instructor's review first and then the peer review to be evaluated.

To evaluate the peer review, follow these steps:

1. Identify the points in the instructor's review in the <evaluation_process> tag. Express the same aspect across different parts as separate points. For each point in the instructor's review:

1) Reprint the text of this point from the instructor's review.

2) Judge whether the content of this point is subjective or objective.

- Objective content includes factual assessments, such as the correctness of the assignment or proofs. - Subjective content includes aspects like clarity or style.

3) Identify the importance of this point:

- Give more weight to critical elements like the correctness of the assignment or proofs.

- Consider subjective elements and minor discrepancies less impactful on the overall score.

4) Extract all relevant text of this point from the peer review.

5) Assess the following aspects:

a. Content: Does the peer review cover the same main topics of this key point? b. Accuracy: Are the peer reviewer's observations and critiques accurate when compared to the instructor's key point? c. Depth: Does the peer review provide an appropriate level of detail and insight?

6) Judge the overall quality of the peer review on this point.

2. According to your evaluation, offer a comprehensive assessment of this peer review in the <assessment> tag, supported by justification.

- highlighting the alignments or misalignments between the peer review and the instructor's review.

- Taking into account both the importance of each key point and the degree of alignment.

3. After the assessment, first provide your reasoning, then assign a score from 0 to 10 based on the rubric, enclosed in the <scoring> tag.

- 0-1: Totally wrong or meaningless review: The review is irrelevant, incoherent, or shows a complete misunderstanding of the material.

- 2-3: Poor review: The review demonstrates significant factual inaccuracies or fails to address essential key points.

- 4-6: Somewhat valuable review: The review contains clear errors or omissions, but partially aligns with the instructor's review on some important points.

- 7-9: Good review: The review largely aligns with the instructor's review on key points, with only minor inaccuracies or omissions.

- 10: Exceptional review: The review is highly consistent with the instructor's on both content and reasoning, with minimal flaws.

4. Output your final score again in the <final_score> tag, with only the number.

Present your final evaluation in the following format:

<evaluation_process>Point 1: [Description]

- Instructor's review: [Reprint text of this point from the instructor's review]

- Objective/subjective: [Reasoning first to judge whether the content of this point is subjective or objective]

- Importance: [Reasoning first to identify the importance of this point]

- Peer review: [Extract all relevant text of this point from the peer review]

- Assessment: [Assess the content, accuracy, and depth in detail]

- Quality: [Judge the quality of the peer review in relation to this point]

Point 2: [Description] ...</evaluation_process>

<assessment>[Your comprehensive assessment of this peer review]</assessment><scoring>[Your reasoning and the score for the peer review based on the rubric]</scoring><final_score>[Output the final score]</final_score>

Here is your input:

<instructor_review>INSTRUCTOR_REVIEW</instructor_review>

<peer_review>PEER_REVIEW</peer_review>

# B ADDITIONAL RESULTS

This section presents experimental results that are omitted from the main text.

## B.1 LLM-JUDGE SCORES USING GPT

In our primary experiments, we obtain LLM-judge scores by querying the gemini-2.5-flash-preview-04-17 model to assess each peer review against its corresponding instructor review, according to a predefined scoring rubric.

To evaluate the robustness of this approach, we repeated the procedure using GPT-4.1 with the same prompt, thereby constructing a GPT-based LLM-judge. The resulting scores are shown in Figure 5. LLM-Judge with GPT shows a lower consistency with the instructor score.

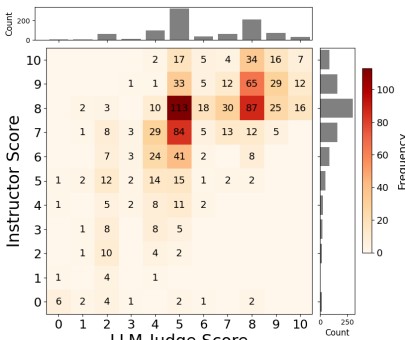

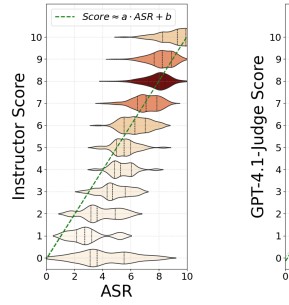 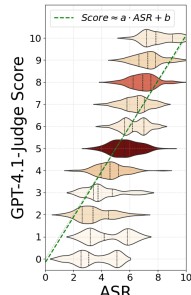

Figure 6: Reference Scores vs. ASR. Left: instructor score vs. ASR aligned with instructor score. Right: LLM-Judge score vs. ASR aligned with LLM-Judge score. The green line represents the linear regression fitting reference score from ASR, which is nearly the identity function in both plots.

Figure 5: Joint distribution (instructor score vs. LLM-Judge score using GPT-4.1).

Figure 6 presents the same linear regression fitting the reference score from our ASR. The regression line remains almost identical.

## C  CASE DEMONSTRATION

We present an example of ASR in this section. Figure 7 visualizes the single-dimensional scoring rules. The homework assignment is on asymptotic analysis and is divided into three parts $A, B, C$, each corresponding to the asymptotic relationship between two functions. For each dimension, we plot the V-shape scoring rule for this dimension.

From the plot, we can observe the dimensions that are not important for scoring, where the scoring line is almost linear, meaning the score does not depend on the report but only on the state. For example, we observe that the dimensions for clarity are less important, e.g., "part A details are clear" and "submission well-structured".

We also identify important dimensions, where the two linear scoring lines form a more strongly convex function. We observe that summary points on details related to overall correctness are more important, e.g., "Algorithm logic is correct", "solution omits details", Dim 4 "Part B is correct", and "Part A is sound".

In general, we observe that our ASR when learning Instructor Score assign more convex V-shape scoring rule to the content that is commonly considered to be more important.

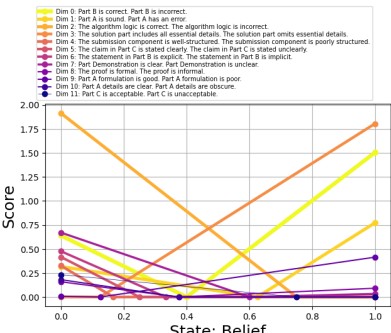

Figure 7: The visualization of ASR on one assignment in the algorithm class using instructor score as the reference. The score of $r = \bot$ for each dimension has been shifted to zero.

