# OpenReview forum: "Aligned Textual Scoring Rule"
_ICLR.cc/2026/Conference — Submitted to ICLR 2026_

### Official Review · Reviewer_DHrE · 2025-10-30

**Soundness:** 1
**Presentation:** 1
**Contribution:** 1
**Rating:** 2
**Confidence:** 5

**Summary:**

The paper builds on textual elicitation via LLM oracles and proposes an Aligned Scoring Rule (ASR): optimize (via convex programming) over proper scoring rules to minimize MSE against a reference score (instructor or LLM-judge), aiming to keep properness while improving human alignment. It relies on the ElicitationGPT reduction (summarization + QA oracle) and evaluates on peer-grading data. While the formal scaffolding (proper rules, separate aggregation, convex objective) is tidy, the narrative arc abruptly ends with appendices: there is no conclusion/discussion section, and multiple central claims are under-supported.

**Strengths:**

Practical intent: aligning a provably proper rule to human/LLM reference scores for peer grading is useful, with simple metrics (MSE / Pearson / Spearman).

**Weaknesses:**

- No Conclusion / Missing Discussion

The manuscript lacks a conclusion (and broader limitations/future-work discussion). For a mechanism-design paper making alignment claims, this omission is severe—readers are left without synthesized takeaways or scope boundaries.

- Over-general claim vs. narrow evaluation

The paper positions ASR as a general textual scoring alignment method, yet experiments are confined to peer grading in two undergraduate algorithm classes (22 assignments, 6–8 submissions each). For such a broad claim, 1 domain / 1 application family is insufficient (no summarization, coding, reasoning competitions, multi-hop QA, etc.).

- Mechanism rationale under-explained

While ASR enforces properness and minimizes MSE to a reference, there is no theoretical argument (or stress test) explaining why MSE alignment to potentially noisy or biased references (instructor / LLM-judge) yields better truthful textual elicitation in the wild. E.g., no analysis of error propagation from oracles, sample complexity, or generalization beyond the training clusters.

- Baselines and diagnostics are thin for the claim

Comparisons cover best-constant and non-aligned ElicitationGPT (AV/MV), plus correlations against instructor/LLM-judge. Missing are: (a) stronger parametric aligners (e.g., small finetuned judger trained on the same reference), (b) adversarial/strategic probes that the paper’s properness motivation emphasizes, and (c) deeper ablations on oracle quality and cluster granularity.

**Questions:**

N/A

---

### Official Review · Reviewer_ddre · 2025-11-09

**Soundness:** 2
**Presentation:** 1
**Contribution:** 1
**Rating:** 0
**Confidence:** 4

**Summary:**

This paper violates the paper guidelines, and therefore i recommend rejection.

**Strengths:**

I have not checked the paper, as the paper has formatting errors and violates the ICLR guidelines.

**Weaknesses:**

I have not checked the paper, as the paper has formatting errors and violates the ICLR guidelines.

**Questions:**

Not applicable.

---

### Official Review · Reviewer_2wqj · 2025-11-10

**Soundness:** 2
**Presentation:** 2
**Contribution:** 2
**Rating:** 0
**Confidence:** 1

**Summary:**

I was told to write a placeholder review here, since this paper is getting desk rejected due to not following the template.

**Strengths:**

I was told to write a placeholder review here, since this paper is getting desk rejected due to not following the template.

**Weaknesses:**

I was told to write a placeholder review here, since this paper is getting desk rejected due to not following the template.

**Questions:**

I was told to write a placeholder review here, since this paper is getting desk rejected due to not following the template.

---

### Official Review · Reviewer_VY1n · 2025-11-10

**Soundness:** 2
**Presentation:** 2
**Contribution:** 2
**Rating:** 0
**Confidence:** 4

**Summary:**

The paper has no "Under review as a conference paper at ICLR 2026" above each page. Therefore, I suggest that the paper should be desk rejected.

**Strengths:**

NA

**Weaknesses:**

NA

**Questions:**

NA

---

### Meta-Review · Area_Chair_JmaM · 2026-01-06

**Summary:**

There is agreement that the paper should be rejected because it doesn't use the right template and violates the submission guidelines. There was no reply by the authors during the rebuttal phase.

**Reviewer Concerns:**

See above

**Reviewer Scores:**

See above

---

### Decision · Program_Chairs · 2026-01-26

Reject